# Zearalenone and Its Metabolites—General Overview, Occurrence, and Toxicity

**DOI:** 10.3390/toxins13010035

**Published:** 2021-01-06

**Authors:** Karolina Ropejko, Magdalena Twarużek

**Affiliations:** Department of Physiology and Toxicology, Faculty of Biological Sciences, Kazimierz Wielki University, Chodkiewicza 30, 85-064 Bydgoszcz, Poland; kararop@ukw.edu.pl

**Keywords:** mycotoxin, zearalenone, contamination, toxicity, public health

## Abstract

Mycotoxins are secondary metabolites of filamentous fungi and represent one of the most common groups of food contaminants with low molecular weight. These toxins are considered common and can affect the food chain at various stages of production, harvesting, storage and processing. Zearalenone is one of over 400 detected mycotoxins and produced by fungi of the genus *Fusarium*; it mainly has estrogenic effects on various organisms. Contaminated products can lead to huge economic losses and pose risks to animals and humans. In this review, we systemize information on zearalenone and its major metabolites.

## 1. Introduction

Zearalenone (ZEN) is a mycotoxin produced by fungi of the genus *Fusarium* [1], mainly *F. graminearum*, *F. culmorum*, *F. cerealis*, *F. equiseti*, *F. crookwellense*, *F. semitectum* [2], *F. verticillioides, F. sporotrichioides, F. oxysporum* [3] and *F. acuminatum* [4]. Fungi especially produce ZEN in temperate and warmer climates [5]. Zearalenone has the general formula C_18_H_22_O_5_ [6] (Figure 1) and is a 6-(10-hydroxy-6-oxy-trans-1-undecenyl-beta-resorcylic acid lactone) [6]. It was isolated, for the first time (as F-2), from maize inoculated with Fusarium [7].

The name “zearalenone” is derived from the combination of the terms maize (Z*ea mays*)—“zea”, resorcylic acid lactone—“ral”, —“en” for the presence of a double-bond, and “one” for the ketone group [6]; ZEN is a non-steroidal estrogen mycotoxin [8] biosynthesized via the polyketide pathway [9].

The structure of ZEN is similar to that of naturally occurring estrogens such as estradiol, estrone, estriol [10], 7β-estradiol [11], and 17-β-estradiol [12]. It has a molar mass of 318.364 g/mol and is a weakly polar compound in the form of white crystals, with blue-green fluorescence at 360 nm excitation and green fluorescence at 260 nm UV excitation [13]. The melting point of ZEN is 164–165 °C. Although it is insoluble in water [14,15], it dissolves well in various alkaline solutions such as benzene, acetonitrile, acetone, or alcohols [16]. Zearalenone is thermostable [17], and is not degraded by processing such as milling, extrusion, storage, or heating [10]. This mycotoxin accumulates in grains mainly before the harvest, but also after harvesting under poor storage conditions [5].

Suitable conditions for the production of ZEN by fungi are characterized by temperatures between 20 and 25 °C and humidity above 20%, when ZEN can be generated within 3 weeks. However, when fungi are exposed to stress and low temperatures of 8–15 °C, they will produce ZEN within a few weeks [3]. Research has shown that high levels of zearalenone in grains are frequently found in countries with a warm and wet climate [18].

Zearalenone is metabolized in the intestinal cells and has two main metabolites: α-zearalenol (α-ZEL) (a synthetic form of zearalenone) and β-zearalenol (β-ZEL); they are formed via the reduction of ZEN [9,19]. Other forms of zearalenone are α-zearalanol (α-ZAL) and β-zearalanol (β-ZAL) [20]. In its metabolized form, it can be conjugated with glucuronic acid [10]. Due to the double bond in the lactone ring (C_11_ and C_12_), ZEN can exist as two isomers: trans and cis, of which the cis form has a greater affinity for estrogen receptors [21]. Of the metabolites, α-ZEL has increased estrogenic activity compared to α-ZAL and ZEN produced by pig liver microsomes, while chicken microsomes produce the highest amounts of β-ZEL, which has, however, a lower estrogenic activity [3,22], but is the most frequently detected metabolite in cattle [23,24]. Hydroxylation of ZEN to α-ZEL is an activation process, whereas the production of β-ZEL is a deactivation process [3]. Böswald et al. [25] investigated the ability of certain yeast strains to metabolize ZEN and showed that ZEN, α-ZEL, and β-ZEL were reduced by *Candida, Hansenula, Pichia*, and *Saccharomyces* species. The fungal species *Clonostachys rosea* has the ability to metabolize the ester bond in the ZEN lactone ring, which reduces its estrogenic activity [26]. Infected plants can metabolize fungal toxins mainly by forming glucose conjugates, and studies have shown that ZEN can be converted to zearalenone-14-O-β-glucoside, which does not interact with the human estrogen receptor in vitro [18]. Based on results, adsorption of ZEN can occur on the hydrophobic talc surface, which is more effective than the hydrophilic diatomaceous earth surface. This makes the use of talc as a sorbent a promising method of ZEN decontamination [27].

## 2. The Occurrence of ZEN in Food

Due to its toxicity, the presence of ZEN in food has been widely studied. The European Commission has specified the maximum standards of ZEN in selected food products (Commission Regulation (EC) No. 1881/2006 and Commission Recommendation No. 2006/576/EC, as amended) [28,29] (Table 1).

Zearalenone has been detected frequently in different cereals, such as wheat, barley, maize, sorghum, rye [2,5], rice [2], corn silage [3], sesame seed, hay [10], flour, malt, soybeans, beer [30], and corn oil [26].

It can also occur in grain-based products such as grains for human consumption, baked goods, pasta breakfast cereals [5], and bread [31]. When cows consume foods contaminated with ZEN, it can be detected in their milk [32,33], thereby reaching the human food chain.

The result of research on the presence of ZEN in food conducted by scientists from around the world are presented in the tables below (Table 2).

Several studies have found ZEN metabolites in various food items (Table 3).

The data presented in Table 2 refer to presence of ZEN in food. On their example, the following conclusions can be drawn: the most contaminated samples are samples of maize, raw maize, corn, beans, grains and feed mixtures for fattening pigs (over 75% of positive samples in the described examples), while the least contaminated are samples of wheat, peas, barley, cow’s milk-based infant formula and beer (up to 15% of positive samples in the described examples). The highest levels of ZEN were found in samples of corn, corn grains, fibrous feed, feed mixtures for fattening pigs and fish feed. This confirms that grains and feeding stuff are the most exposed to the presence of ZEN. However, it should be remembered that these are data selected from many publications by authors from around the world. The data presented in Table 3 refer to presence of ZEN metabolites in food products. On their basis it can be concluded that the ZEN metabolites are not common in food as ZEN itself. The most common was α-ZEL in the chicken heart and chicken gizzard samples, nevertheless, the levels detected were relatively low—mean 3.60–4.01 µg/kg. The highest level of α-ZEL was found in the fish feed—188.4 ng/mL.

## 3. The Occurrence of ZEN in Body Fluids

ZEN and its metabolites are absorbed by the body when ingested with food. For this reason, it can appear in biological fluids such as blood, urine and milk (including women breast milk). Research on biological fluids of various species is carried out in many countries around the world. The table below (Table 4) presents the results of research by scientists from individual countries. ZEN occurrence in body fluids indicates the presence of ZEN in a body. This is disadvantageous because of the damage to the organism that ZEN causes.

Table 4 shows examples of the occurrence of ZEN in body fluids such as serum, milk and urine. Of all examples presented, the highest level of ZEN was found in urine of pig’s samples (male—350 µg/L and female—390 µg/L), while in humans it was in the urine of men from Germany—100 ng/L. High levels of ZEN have also been found in the urine of breastfed (784 ng/L) and non-exclusively breastfed infants (678 ng/L). This may indicate that ZEN is metabolized more slowly in infants than in adults.

Mauro et al. in 2018 [63] conducted a study whose results showed that ZEN is present in the serum of obese women. This may be related to meat consumption and body mass index. The level of ZEN however, was lower than that of women of normal weight. 0.405 ± 0.403 ng/mL and 0.711 ± 0.412 ng/mL, respectively. In addition, the same study showed that the mean values of conjugated metabolites of ZEN in premenopausal women were higher than in postmenopausal women—1.40 ± 0.645 and 1166 ± 1007 ng/mL, respectively.

In the last few years, the influence of ZEN and its metabolites on human health has been increasingly studied. In 2002, Pillay et al. [64] Conducted a study of the serum of patients with breast cancer, cervical cancer, other gynecological diagnoses and healthy. The research did not show any significant changes between the presence of ZEN and its metabolites in the tested samples. Mean ± SD ZEN values ranged between 0.457 ± 1.06 µg/mL, 0.381 ± 0.82 µg/mL, 0.200 ± 0.38 µg/mL, 0.346 ± 0.51 µg/mL in breast cancer, cervical cancer, other gynecological diagnoses and healthy samples respectively. Mean ± SD α-ZEL values ranged between 0.193 ± 0.50 µg/mL, 0.154 ± 0.26 µg/mL, 0.070 ± 0.16 µg/mL, and 0.378 ± 0.89 µg/mL in breast cancer, cervical cancer, other gynecological diagnoses and healthy samples respectively, also mean ± SD β-ZEL values ranged between 0.233 ± 0.69 µg/mL, 0.707 ± 1.51 µg/mL, 0.215 ± 0.60 µg/mL, 0.110 ± 0.51 µg/mL in breast cancer, cervical cancer, other gynecological diagnoses and healthy samples respectively. A similar study was conducted by Fleck et al. [65]. Their results showed the presence of ZEN in only 1 out of 11 urine samples of pregnant women with value to the limit of quantification.

Another study was conducted in 2017 by De Santis et al. [66]. The authors investigated the possible relationship between the occurrence of ZEN in the body and autistic disorders. Urine and serum samples of children with autism were examined, the maximum level of ZEN was 6.5 and 3.9 ng/mL, respectively as well as urine and serum samples of their siblings where the maximum ZEN level was 2.8 and 1.2 ng/mL, respectively. These results suggest that patients with autistic disorder have significantly more mycotoxin from body fluids than their healthy siblings who should have similar food habits.

Moreover, Tassis et al. [67] carried out a boar semen analysis. The authors showed that ZEN negatively affects various sperm parameters such as sperm viability and motility.

## 4. The Impact of ZEN on Organisms

Zearalenone is a mycotoxin with immunotoxic [9], hepatotoxic [9], and xenogenic effects [68]. The activity of ZEN in living organisms depends on the immune status of the organism and the state of the reproductive system (adolescence or pregnancy stage) [69]. In the liver, ZEN induces histopathological changes, with the subsequent development of liver cancer [70]; according to Rai et al. [22], the liver is the major organ of ZEN distribution. In the case of liver injury, ZEN can cause an increase in serum transaminases and bilirubin levels in rodents [31]; in addition, it can lead to weight loss in rats [71] and fish [72].

Zearalenone has hematotoxic effects by disturbing blood coagulation and modifying blood parameters [2,22,30]. Studies have shown that in the serum of mice treated with ZEN, the levels of ALT (Alanine Aminotransferase), ALP (Alkaline Phosphatase), and AST (Aspartate Aminotransferase) were increased, while those of total protein and albumin were decreased [22]. In studies conducted in rats, an increase in hematocrit and MCV (mean corpuscular volume) index was observed, while the number of red blood cells remained unchanged; the number of platelets was significantly decreased and that of white blood cells was increased. The same study also showed that the blood creatinine value was decreased in the samples with ZEN [73]. Zwierzchowski et al. [74], in a study on gilts that received small doses of ZEN orally, showed that after the first administration of the toxin, its concentration in the blood serum was high; however, after administration of the same dose in the following days, its level decreased (until day 4) and then increased again.

Zearalenone is a mycotoxin with strong estrogenic [13,75,76] and anabolic effects [75,76]. One of the metabolites of ZEN, α-ZAL, is used as a growth promoter due to its anabolic activity [23]. Zearalenone and its derivatives show estrogenic effects in various animal species. In humans, ZEN can bind to alpha and beta estrogen receptors and disrupt the functioning of the endocrine system [18]. The species most sensitive to the effects of ZEN are pigs [3,8,20,22] and ruminants [20], while the most resistant ones are birds [20], such as chickens [31] and poultry [77]. The estrogenic effects of ZEN include fertility disorders (infertility or reduced fertility), vaginal prolapse, vulvar swelling and breast enlargement in females, feminization of testicular atrophy, and enlargement of the mammary glands in males in various animal species [78]. It can also cause enlargement of the uterine, increased incidence of pseudopregnancy, decreased libido, stillbirths, and small litters [3]. In female pigs, redness and swelling of the vulva, enlargement of the uterus, cyst formation on the ovaries, and enlargement of the mammary glands have been observed, whereas in male pigs, testicular atrophy and reduced sperm concentration are common [79]. Zearalenone inhibits the secretion of steroid hormones, interferes with the estrogen response in the pre-ovulatory phase, and inhibits follicle maturation in mammals [24]. Higher concentrations of ZEN cause permanent estrus, pseudo-pregnancy, and infertility in gilts [80]. In cows, symptoms of ZEN actions are swollen vulva, disturbances in estrus cycles, infertility, inflammation of the uterus and mammary gland, miscarriages, placental retention, and vaginitis [81]; ZEN is also responsible for the hyperestrogenic syndrome [24,82]. Newborn female mice that received ZEN orally showed altered oocyte development and folliculogenesis later in life [24]. In humans, ZEN causes premature puberty [83]. In pregnant women, long-term exposure to ZEN via food may result in decreased embryo survival and reduced fetal weight, as well as decreased milk production. It is also assumed that ZEN can change uterine tissue morphology and cause a decrease in LH and progesterone levels [2]. In men, ZEN reduces the number of sperm and their viability [84]; it can also impede spermatogenesis [2].

Studies on the estrogenic effect of ZEN and its modified forms have been carried out in zebrafish (model fish species), showing that ZEN passively crosses the cell membrane and binds to ER receptors. The ZEN receptor complex is rapidly transported to the nucleus, where it binds to estrogen-responsive elements, resulting in gene transcription [85]. Pietsch et al. [72] fed carp (*Cyprinus carpio* L.) with ZEN-contaminated feed and showed that the estrogenic activity in these animals was not increased, indicating that ZEN is rapidly metabolized in carp.

According to Gil-Serna et al. [2], ZEN is also genotoxic and can form DNA adducts in vitro. Further, it causes DNA fragmentation, micronucleus formation, chromosomal aberration, cell proliferation, and cell apoptosis [22]. Research shows that ZEN and β-ZEL can mimic the ability of 17-β-estradiol to stimulate estrogen receptor transcriptional activity [86]. The International Agency for Research on Cancer (IARC) has classified ZEN as a Group 3 substance (not carcinogenic to humans) [15]. Zearalenone cytotoxicity can manifest by apoptosis in the germ cells of male rats [87].

The WHO/FAO determined the lowest observed adverse effect level (LOAEL) of ZEN at 200 µg/kg bw/day study in a 15-day pig study [88], 56 µg/kg bw/day for sheep, 17.6 µg/kg bw/day for piglets, 200 µg/kg bw/day for gilts, and 20 µg/kg bw/day for dogs [85]. The no effect level (NOEL) was 40 µg/kg bw/day for pigs [30,31], 9200 µg for mice [89], 28 lg/kg bw/day for sheep [85], 100 μg/kg bw for rats [10,30], 10.4 µg/kg bw/day for piglets, and 40 µg/kg bw/day for gilts [85].

Obremski et al. investigated the effect of LOAEL doses on gilts and showed that an orally administered dose of 200 µg/kg (the LOAEL dose) caused mild symptoms of hyperestrogenism in sexually immature gilts on the fourth day after toxin administration, whereas a dose twice as high (400 µg/kg) resulted in more pronounced symptoms of hyperestrogenism on the third day after oral administration of the toxin [90].

The oral LD_50_ ZEN dose for mice, rats, and guinea pigs is above 2000 mg/kg bw [91], and the median toxic dose (TD_50_) was established at 20,000 µg for mice [89]. The EFSA Panel on Contaminants in the Food Chain stated a tolerable daily intake (TDI) for ZEN of 0.25 μg/kg bw [5,18].

Table 5 shows the various parameters of ZEN.

## 5. Toxicokinetics of ZEN

The toxicokinetics of ZEN mainly include issues such as the rate at which it can enter the body, absorption, distribution, metabolism and excretion. The main way for ZEN to enter organisms is through its consumption with contaminated food. In organisms, it can undergo structural changes through the intestinal microflora. These changes lead to the production of various ZEN metabolites [22].

After oral administration, ZEN is rapidly absorbed. In the intestinal walls of monogastric animals and the human gastrointestinal tract, ZEN is metabolized by enterocytes to the major metabolites α- and β-ZEL and α- and β-ZAL, followed by biotransformation [31,92] via two pathways. The first is based on hydroxylation, leading to the formation of α- and β-ZEL when catalyzed by 3α- and 3β-hydroxysteroid dehydrogenases (HSD). The α form has a greater affinity for estrogen receptors and is therefore more toxic than ZEN, while the β form has a lower affinity for these receptors, making it practically harmLess. The second biotransformation pathway relies on uridine-5-diphospho-glucuronosyltransferase (UDPGT)-catalyzed conjugation of ZEN and its metabolites with glucuronic acid. In humans, ZEN biotransformation occurs in the liver, lungs, kidneys, and intestines [9,20,84]. Nevertheless, in human organisms, it is mainly in the liver that ZEN is converted into α and β isomers via microsomes. In it, the metabolizing ZEN is through monohydroxylation via cytochrome P450 (CYP) [22].

After oral administration, ZEN is rapidly absorbed. In pigs, it has been detected in plasma less than 30 min after starting feeding. It is deposited in the reproductive tissues, adipose tissue, and testicular cells [3], as well as in the kidney cells [5]. Its half-life in pigs is approximately 86 h [3,22], and in these animals, absorption from the gastrointestinal tract occurs to 80–85% [5]. In other organisms, ZEN and its metabolites have a short half-life of less than 24 h [93] and are mainly excreted in the bile [22,70], feces [3,10,22], and urine [3,9,22] after 72 h [3]. Metabolism includes Phase I of the reduction reaction and Phase II of the glucuronidation or sulfonation reaction [12]. Metabolism of phase I reduce keto group at C-6′ resulting α-ZEL or β-ZEL. Following reduction of the double bond C11-C12 leads to α-ZAL or β-ZAL. Studies show that the reduction of the ketone group is catalyzed by HSD [31]. Hepatic biotransformation may be influenced by species differences and related ZEN sensitivities. The largest amounts of α-ZEL, which has the highest estrogenic activity, are produced by the liver microsomes of pigs, while the microsomes of chickens, which produce the most β-ZEL, which has the lowest estrogenic activity [3].

ZEN and its metabolites can interact with the cytoplasmic receptor it binds to 17β-estradiol and transfer receptors to the nucleus, where RNA simulation leads to protein synthesis which is the reason why the estrogenic symptoms occur [3].

In conclusion, ZEN and its metabolites are eliminated relatively slowly from the tissues by enterohepatic circulation. The carry-over to milk is quite low, confirming that human exposure to food of animal origin is significantly lower than direct exposure through the use of defective feed and grains [31].

## 6. Conclusions

Zearalenone is the main mycotoxin produced by *Fusarium* and can negatively affect most species. It causes various changes and disorders related to the reproductive system, generating considerable economic losses. Regarding the toxicity of zearalenone and its metabolites, they pose a potential risk to mammals, especially when exposed to high doses over prolonged periods. Consuming excessive amounts of mycotoxins can cause poisoning, the so-called “mycotoxicosis”, posing a considerable threat for animals and humans. In this review, we present the various effects of zearalenone and its metabolites. Based on the ubiquitous occurrence of these compounds, it is crucial to develop methods of decontamination and to impede the production of zearalenone.

## Figures and Tables

**Figure 1 toxins-13-00035-f001:**
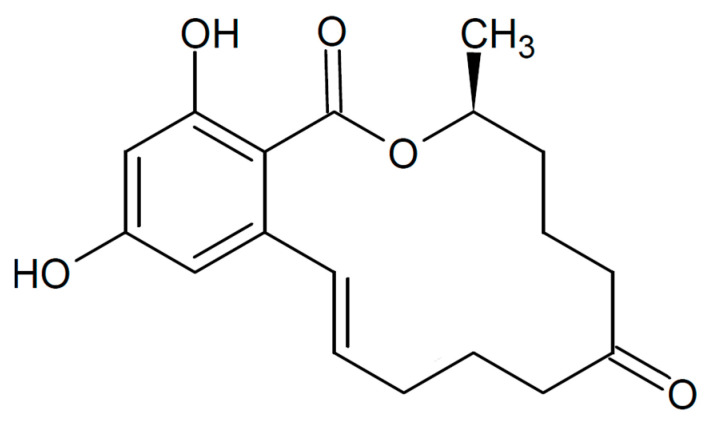
Structural formula of zearalenone.

**Table 1 toxins-13-00035-t001:** Maximum standards of zearalenone in selected food products (Commission Regulation (EC) No. 1881/2006 and Commission Recommendation No. 2006/576/EC, as amended).

Product	Highest Permissible Value [μg/kg]
Unprocessed cereals other than maize	100
Unprocessed maize	350
Cereals intended for direct human consumption, cereal flour, bran as end product marketed for direct human consumption and germ	75
Refined corn oil	400
Maize intended for direct human consumption, maize snacks, and maize-based breakfast cereals	100
Bread (including small bakery wares), cakes, biscuits, cereal snacks, and breakfast cereals, excluding maize snacks and maize based breakfast cereals	50
Processed cereal-based foods (excluding processed maize-based foods) and baby foods for infants and young children	20
Processed corn-based foods for infants and young children	20
Compound feed for piglets, gilts, puppies, kittens, dogs, and cats intended for reproduction	0.1
Compound feed for adult dogs and cats other than those intended for reproduction	0.2
Compound feed for sows and porkers	0.25
Compound feed for calves, dairy cattle, sheep (including lambs), and goats (including goatlings)	0.5

**Table 2 toxins-13-00035-t002:** The presence of zearalenone in different food items.

Country	Products	% of Positive Samples (Number of Samples)	Results	References
Croatia	Maize	80% (12/15)	range 0.62–3.2 μg/kg	[34]
Argentina	Raw maize	100% (26/26)	mean 15 μg/kg, maximum 42 μg/kg	[35]
Bulgaria	Maize	21.1% (4/19)	mean 80.6 μg/kg, maximum 148 μg/kg	[36]
Morocco	Corn	15% (3/20)	mean 14 μg/kg, maximum 17 μg/kg	[37]
Germany	Corn	85% (35/41)	mean 48 μg/kg, maximum 860 μg/kg	[38]
Argentina	Corn grains	36% (21/58)	maximum 1560 μg/kg	[39]
Spain	Corn snacks	23.6% (17/72)	maximum 22.8 μg/kg	[40]
Germany	Wheat	63% (26/41)	mean 15 μg/kg	[38]
Bulgaria	Wheat	1.9% (1/54)	mean 10 μg/kg, maximum 10 μg/kg	[36]
Germany	Oats	24% (4/17)	mean 21 μg/kg	[38]
Germany	Hay	42% (12/28)	mean 24 μg/kg, maximum 115 μg/kg	[38]
Germany	Peas	0%	-	[38]
South Korea	Beans	100% (1/1)	maximum 15 µg/kg	[15]
South Korea	Grains	77% (17/22)	maximum 277 µg/kg	[15]
Bulgaria	Barley	11.1% (2/18)	mean 29 μg/kg, maximum 36.6 μg/kg	[36]
Germany	Soya meal	69% (9/13)	mean 51 μg/kg, maximum 211 μg/kg	[38]
South Korea	Fibrous feed	50% (4/8)	maximum 1315 µg/kg	[15]
South Korea	Food byproducts	62% (8/13)	maximum 176 µg/kg	[15]
Croatia	Feed mixtures for fattening pigs	93.3% (28/30)	range 8.93–866 μg/kg	[41]
Kenya	Fish feed	40% (31/78)	range from < 38.0–757.9 ng/mL	[42]
China	Eggs	44% (32/72)	range between 0.30–418 µg/kg	[43]
Pakistan	Eggs	45% (18/40)	mean ± SD 2.23 ± 0.51 μg/kg	[44]
Pakistan	Chicken meat	52% (60/115)	mean ± SD 2.01 ± 0.90 μg/kg	[44]
Iran	Buffalo meat	41.42% (29/70)	range from 0.1–2.5 ng/mL	[45]
Iran	Buffalo liver	68.57% (48/70)	range from 0.1–4.34 ng/mL	[45]
Italy	Cow’s milk-based infant formula	9% (17/185)	maximum 0.76 μg/L	[46]
Pakistan	Bread (corn)	43% (6/14)	mean ± SD 9.45 ± 2.76 μg/kg	[47]
Spain	Sliced bread	43.6% (31/71)	maximum 20.9 μg/kg	[40]
Spain	Beer	11.3% (8/71)	maximum 5.1 μg/kg	[40]
Spain	Pasta	14.3% (10/70)	maximum 5.9 μg/kg	[40]

**Table 3 toxins-13-00035-t003:** Presence of zearalenone metabolites in foods.

ZEN Metabolites	Country	Products	% of Positive Samples (Number of Samples)	Results	References
α-ZEL	Italy	Cow’s milk-based infant formula	26% (49/185)	maximum 12.91 μg/L	[46]
β-ZEL			28% (53/185)	maximum 73.24 μg/L	
α-ZEL	China	Chicken heart	40% (8/20)	mean 3.60 µg/kg	[43]
		Chicken Gizzard	40% (8/20)	mean 4.01 µg/kg	
α-ZEL	Kenya	Fish feed	24% (19/78)	range from < 22.2–288.4 ng/mL	[42]
β-ZEL			33% (26/78)	range from < 16.0–79.8 ng/mL	

**Table 4 toxins-13-00035-t004:** The results of studies of ZEN metabolites found in body fluids.

Country	Body Fluid	% of Positive Samples (Number of Samples)	Results	References
Romania	Pig’s serum	17.3% (9/52)	mean 0.8 ng/mL, maximum 0.96 ng/mL	[48]
Bulgaria	Pig’s serum	50% (5/10)	mean ± SD 0.24 ± 0.12 μg/L	[49]
Bulgaria	Pig’s serum	50% (5/10)	mean ± SD 0.33 ± 0.17μg/L	[49]
Iran	Buffaloes milk	21.42% (15/70)	range between 0.1–3.55 ng/mL	[45]
Spain	Breast milk	37% (13/35)	range between 2.1–14.3 ng/mL	[50]
Italy	Breast milk	100% (47/47)	range between 0.26–1.78 μg/L	[51]
Italy	Breast milk (women with Celiac Disease)	4% (12/275)	range between 2.0–17 ng/mL	[52]
Italy	Breast milk	8% (15/178)	range between 2.0–22 ng/mL	[52]
China	Raw milk	100% (30/30)	mean ± SD 14.9 ± 6.0 ng/kg	[53]
China	Liquid milk	100% (12/12)	mean ± SD 20.5 ± 11.1 ng/kg	[53]
Croatia	Pig’s urine (male)	100% (11/11)	mean ± SD 238 ± 30 µg/L, range between 104–350 µg/L	[54]
Croatia	Pig’s urine (female)	100% (19/19)	mean ± SD 187 ± 27.1 µg/L, range between 22.7–390 µg/L	[54]
Sweden	Pig’s urine	92% (179/195)	mean ± SD 2.44 ± 4.39 ng/mL	[55]
Cameroon	Human urine	3.6% (8/220)	mean 0.97 ng/mL, range between 0.65–5.0 ng/mL	[56]
Nigeria	Human urine	0.8% (1/120)	mean 0.3 µg/L	[57]
Italy	Human urine	100% (52/52)	mean 0.057 ng/mL, maximum 0.120 ng/mL	[58]
Sweden	Human urine	37% (92/252)	mean ± SD 0.09 ± 0.07 ng/mL	[59]
Germany	Male urine (control)	100% (13/13)	mean ± SD 31 ± 23 ng/L, range between 7–90 ng/L	[60]
Germany	Male urine (Mill worker)	100% (12/12)	mean ± SD 42 ± 26 ng/L, range between 4–100 ng/L	[60]
Germany	Female urine (Mill worker)	100% (5/5)	mean ± SD 35 ± 28 ng/L, range between 6–78 ng/L	[60]
Nigeria	Human urine	81.7% (98/120)	mean 0.75 ng/mL, range between 0.03–19.99 ng/mL	[61]
Nigeria	Breastfed infants urine	57% (13/23)	mean 148 ng/L, range between 17–784 ng/L	[62]
Nigeria	Non-exclusively breastfed infants urine	83% (35/42)	mean 140 ng/L, range between 13–678 ng/L	[62]

**Table 5 toxins-13-00035-t005:** Comparison of parameters describing ZEN.

Parameter	Value
LOAEL	200 µg/kg bw/day (15-day pig study)
LOAEL	56 µg/kg bw/day (sheep)
LOAEL	17.6 µg/kg bw/day (piglets)
LOAEL	200 µg/kg bw/day (gilts)
LOAEL	20 µg/kg bw/day (dogs)
NOEL	40 µg/kg bw/day (pigs)
NOEL	9200 µg (mice)
NOEL	100 μg/kg bw (rats)
NOEL	28 µg/kg bw/day (sheep)
NOEL	10.4 µg/kg bw/day (piglets)
NOEL	40 µg/kg bw/day (gilts)
LD_50_	2000 mg/kg (mice, rats, and guinea pigs)
TD_50_	20,000 µg (mice)

## Data Availability

Data sharing not applicable.

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
