# Peer review of "Zearalenone and Its Metabolites—General Overview, Occurrence, and Toxicity"

_toxins, 2021, doi:10.3390/toxins13010035_

Round 1

Reviewer 1 Report

The paper submitted for review is an excellent literature study based on a review of many scientific articles, most of which were published in the last years. The authors touch upon very important issues of mycotoxins – zearalenone - and its major metabolites produced by fungi of the genus Fusarium. Mycotoxins can pose risks to organisms and lead to huge economic losses. Zearalenone can cause poisoning the so-called “mycotoxicosis”, posing a considerable threat for humans and animals.

The manuscript was prepared with care and its content contains a lot of valuable information. It does not raise any scientific or substantive reservations.

It is a clearly written manuscript. All tables and figure are clear, understandable and necessary. The overall quality of the manuscript is good. The references are sufficient and necessary.

The paper needs some editorial corrections. (Check the references in accordance with the journal style). Some comments and recommendations have been included in the text.

I don't feel qualified to review the English used by the authors but I understood well all the text.

I think that the paper can be published in the Toxins after minor revision.

Author Response

Comments: The paper needs some editorial corrections. (Check the references in accordance with the journal style). Some comments and recommendations have been included in the text.

Answers:

  1. In section „Introduction”: in lines 1-3, 6, I used italics in Latin names; in line 8, I wrote the Latin name with a capital letter; in lines 16, 22-23, I changed the degree sign; in lines 39, 41, I changed the citation, as required by the journal style
  2. In section „The occurrence of ZEN in food”: in the table 1 I change the size of first letter in “zearalenone”;
  3. In section “The impact of ZEN on organisms”: in lines 5, 6, 17, 20, 53, 55, 56-57 I changed the citation, as required by the journal style
  4. In section “References”:

in references 7, 8, 17, 22, 25, 35, 38 I used italics in Latin names; in references 10, 11 I change the names of authors; in reference 21 I change the size of first letter in name of journal; in reference 69 I change the title of the publication

Reviewer 2 Report

The authors gave an overview of zearalenone and its metabolites. 

The paper is well written but relatively short for a review paper. Since it refers to only one mycotoxins this could be acceptable, but I would still recommend adding several more pages/sections or to rearrange the existing sections.   

I would recommend to the authors:

1. to reduce the size f figure 1, it is far too big. 

2. to control all Latin names and put them in italic.

3. to make all tables more clear, this is generally too much information in each table, and this makes it hard to follow the content. 

4. to check the references and advise the author's guidelines because some references include a year (in the text), which is not necessary. "According to Gil-Serna et al. (2014),..."

5. Tables should be described more, sections 2 and 3 should not only refer to the tables. You need to add certain explanations about the data you are referring to in the table. 

6. section 5 is too short, add some more text to it. 

I will not comment on the scientific importance of a review paper but it should include all the latest scientific achievements in the topic the authors are referring to. 

Author Response

Comment:

  1. To reduce the size figure 1, it is far too big.

Answer:

  1. In section „Introduction”: I reduced the size of figure 1

Comment:

2. To control all Latin names and put them in italic.

Answer:

  1. In section „Introduction”: in lines 1-3, 6, I used italics in Latin names;
  1. In section “References”: in references 7, 8, 17, 22, 25, 35, 38  I used italics in Latin names

Comment:

3. To make all tables more clear, this is generally too much information in each table, and this makes it hard to follow the content.

Answer: 

  1. In section „The occurrence of ZEN in food”: in tables 2 and 3, I divided the column "results" into two columns: "% of positive samples (number of samples)" and "results"
  1. In section “The occurrence of ZEN in body fluids”: in table 4, I divided the column "results" into two columns: "% of positive samples (number of samples)" and "results"

Comment:

4. to check the references and advise the author's guidelines because some references include a year (in the text), which is not necessary. "According to Gil-Serna et al. (2014),..."

Answer:

  1. In section „Introduction”: in lines 39, 41, I changed the citation, as required by the journal style
  2. 2. In section “The impact of ZEN on organisms”: in lines 5, 6, 17, 20, 53, 55, 56-57 I changed the citation, as required by the journal style

Comment:

5. Tables should be described more, sections 2 and 3 should not only refer to the tables. You need to add certain explanations about the data you are referring to in the table. 

Answer:

  1. In section „The occurrence of ZEN in food”: at the end of this chapter, I added a description of the tables 2 and 3, taking into account the data contained in them
  1. In section “The occurrence of ZEN in body fluids”: at the end of this chapter, I added a description of the table 4, taking into account the data contained in it
    I tried not to duplicate the same data

Comment:

6. section 5 is too short, add some more text to it.

Answer:

  1. In section “Toxicokinetics of ZEN”: I added more information at the chapter (lines 1-5, 16-19, 27-41)

Comment:

7. it should include all the latest scientific achievements in the topic the authors are referring to. I would still recommend adding several more pages/sections or to rearrange the existing sections.  

Answer:

  1. I added the article no 67 in references from 2020. We had 6 articles from 2020 (numbers in the reference list: 12, 17, 26, 42, 62, 68) and 5 from 2019: (numbers in the reference list: 13, 22, 55, 70, 83)

2. I added more information in section “The occurrence of ZEN in body fluids” – lines 39 - 74

Round 2

Reviewer 2 Report

The manuscript has been improved and can be accepted for publication.